# Impact of COVID-19 on acute trauma and orthopaedic referrals and surgery in the UK during the first wave of the pandemic: a multicentre observational study from the COVid Emergency-Related Trauma and orthopaedics (COVERT) Collaborative

Kapil Sugand ![ORCID] ,[1,2] Arash Aframian,[1,2] Chang Park,[2] Khaled M Sarraf,[2] COVERT Collaborative

¹MSk Lab, Imperial College London, London, UK
²Department of Trauma and Orthopaedics, Imperial College Healthcare NHS Trust, London, UK

**Correspondence to**
Dr Kapil Sugand;
ks704@ic.ac.uk

## ABSTRACT

**Objective** This is the first British multicentre study observing the impact of the COVID-19 pandemic on orthopaedic trauma with respect to referrals, operative caseload and mortality during its peak.

**Design** A longitudinal, multicentre, retrospective, observational, cohort study was conducted during the peak 6 weeks of the first wave from 17 March 2020 compared with the same period in 2019.

**Setting** Hospitals from six major urban cities were recruited around the UK, including London.

**Participants** A total of 4840 clinical encounters were initially recorded. 4668 clinical encounters were analysed post-exclusion.

**Primary and secondary outcome measures** Primary outcomes included the number of acute trauma referrals and those undergoing operative intervention, mortality rates and the proportion of patients contracting COVID-19. Secondary outcomes consisted of the mechanism of injury, type of operative intervention and proportion of aerosolising-generating anaesthesia used.

**Results** During the COVID-19 period, there was a 34% reduction in acute orthopaedic trauma referrals compared with 2019 (1792 down to 1183 referrals), and a 29.5% reduction in surgical interventions (993 down to 700 operations). The mortality rate was more than doubled for both risk and odds ratios during the COVID-19 period for all referrals (1.3% vs 3.8%, p=0.0005) and for those undergoing operative intervention (2.2% vs 4.9%, p=0.004). Moreover, mortality due to COVID-19-related complications (vs non-COVID-19 causes) had greater odds by a factor of at least 20 times. For the operative cohort during COVID-19, there was an increase in odds of aerosolising-generating anaesthesia (including those with superimposed regional blocks) by three-quarters, as well as doubled odds of a consultant acting as the primary surgeon.

**Conclusion** Although there was a reduction of acute trauma referrals and those undergoing operative

## Strengths and limitations of this study

► This was the first representative observational study of the UK looking into the impact of COVID-19 pandemic on general trauma and orthopaedic surgical specialty.

► There is a valid comparison between two timeframes, exactly 1 year apart to represent pre-COVID-19 and during COVID-19.

► Other studies thus far have only shed light on local scales or cross-speciality within a shorter timeframe than this study and not necessarily commenting on mortality rates like this study.

► Weaknesses included loss of data points that have been accounted for in the tables (ie, labelled as unknown) which did not affect the final analysis of data points.

► Operations conducted outside the specific study periods will not account for all those operations required such as for hip fractures.

intervention, the mortality rate still more than doubled in odds during the peak of the pandemic compared with the same time interval 1 year ago.

## INTRODUCTION

### The global impact of COVID-19

The COVID-19 pandemic will be remembered as one of the most unprecedented global health crises in modern history. With over 5 million deaths and over 263 million recorded cases globally, the pandemic has had a permanent impact on healthcare at the time of article submission.[1] The viral outbreak was first reported in December 2019 with the first patient hospitalised in

the city of Wuhan, China.[2] By mid-March, the outbreak affected over 190 countries with over 450 000 cases and over 20 000 deaths, thus being declared a pandemic and a global public health emergency by the WHO.[3] On 24 January 2020, Europe reported its first case followed by a case in the UK 5 days later.[4]

## The British response to the pandemic

The English government responded by implementing social distancing measures on 17 March 2020 in an attempt to reduce the rate of transmission and therefore the demands on the National Health Service (NHS).[5] This was followed a week later by more stringent measures, commonly referred to as a societal 'lockdown'.[6] As of 23 March 2020, all members of the public were required to stay at home. The NHS has also been deeply affected by the strain imposed by the virus as the healthcare infrastructure has had to evolve to cope with the overwhelming and unexpected pressures on staff, resources and finances. There has been a complete renovation of emergency medicine and orthopaedic services to manage musculoskeletal disease and trauma. In response to the NHS emergency declaration,[7] the Royal Colleges of Surgeons[8] and the British Orthopaedic Association[9] both issued statements and guidelines for delivering emergency trauma and orthopaedic care during the COVID-19 outbreak. The lockdown to limit the spread of the virus has had an unforeseen effect in significantly reducing the acute trauma workload described in several single-centre studies.[10–13] There has, however, not been a British multicentre reflection of the impact of the COVID-19 pandemic on the orthopaedic workload and its potential impact on patient mortality.

## AIM

To observe the impact of COVID-19 on trauma and orthopaedic acute referrals, operative casemix and mortality rates during the peak 6 weeks of the first wave of the pandemic compared with the same time interval in 2019.

## ALTERNATIVE HYPOTHESIS

When comparing both years, there would be a difference in the prevalence of acute orthopaedic referrals, orthopaedic trauma casemix and aerosol-generating anaesthetic procedures due to social distancing/lockdown. Mortality rates and survival probabilities were also hypothesised to differ due to the first COVID-19 outbreak.

## METHODS
### Study design

This is the first multicentre longitudinal observational study observing patients who were acutely referred to the trauma and orthopaedic departments as well as those operated on within the same 6-week interval comparing 2019–2020 for general orthopaedic trauma.

## Setting

Seven principal NHS hospital trusts contributed data from six major urban cities including London, Gateshead, Middlesbrough, Dartford, Newport and Reading.

## Patient sampling

All acute referrals, operative notes, inpatient medical records and discharge summaries were accessed using electronic patient databases at each contributing hospital trust.

## Study period

The 6-week study period was from the start of social distancing on 17 March 2020 to 28 April 2020 which encompassed the national lockdown measures instigated on 23 March 2020. This period was considered the peak 6 weeks of the epidemic in the UK as outlined by the recorded mortality rates and R-values published by the Office of National Statistics.[14] This time period was compared with the same 6-week interval from 19 March to 30 April 2019 (ie, prior to any COVID-19-related measures) to compare the impact of the pandemic 1 year apart.

## Outcomes/objectives

Primary outcomes included the number of acute trauma referrals and those undergoing operative intervention, postoperative complications, mortality rates and the proportion of patients contracting COVID-19. Secondary outcomes consisted of the mechanism of injury, type of operative intervention and proportion of aerosolising-generating anaesthesia used.

## Inclusion criteria

All acute orthopaedic trauma referrals presenting to the emergency department during the intervals 1 year apart were included. All orthopaedic trauma cases that required an operation, including those from acute orthopaedic trauma referrals, within the intervals 1 year apart. Those patients listed for an operation due to orthopaedic trauma prior to time period of data collection were included in the final analysis. We adhered to Strengthening the Reporting of Observational Studies in Epidemiology guidelines for observational studies.

## Exclusion criteria

Any cases being referred internally from other specialties for trauma and orthopaedic advice and input, as well as referrals from any external centre asking for tertiary advice were excluded from further analysis. Any patients with postoperative complications arising from the period prior to the data collection were excluded. For operative trauma cases, those undergoing spinal procedures were excluded as these are jointly treated by neurosurgery in most hospitals. All non-urgent semi-elective procedures were excluded from analysis as well, as they would inaccurately assess the impact of any social distancing measures on the trauma workload. Routine elective orthopaedic cases were excluded.

**Table 1** Data points for acute referrals and operative casemix

| | | | |
|---|---|---|---|
| Age (years) | Gender (male/female by birth) | ASA (1–5) | Date of injury/presentation |
| Injury | Mechanism of injury | Open versus closed fracture | Trauma call (yes/no) |
| Operative procedure | Anaesthetic technique (AGP vs non-AGP) | Seniority of surgeon (consultants vs trainees) | Comorbidities |
| Six-week mortality | Post-op complications | Surgery time since admission (hours) | COVID-19 status (from PCR swabs) |

AGP, aerosolising-generating procedure; ASA, American Society of Anesthesiologists.

## Data points

Demographics including age, sex and American Society of Anesthesiologists (ASA) grades were recorded for all patients. Injury characteristics were recorded, including the anatomical location and if the injury was open or closed. The mechanism of injury was categorised and whether the patient was referred as a trauma call. The nature of the operative procedures and the anaesthetic techniques were recorded. Patients undergoing multiple procedures were recorded for every episode where they were taken to theatre. Six-week mortality rate was recorded as well as the COVID-19 status of any symptomatic patients or suspected cases. Data points were divided into acute referrals and operative casemix as seen in table 1.

► *Anaesthetic techniques*: This was divided into anaesthetic aerosolising-generating procedures (AGPs) which consisted of any intubation (including laryngeal mask airway and endotracheal intubation) for a general anaesthetic. All other anaesthetic techniques including regional and local anaesthetics were deemed as non-AGPs.

► *COVID-19 status*: At the time, COVID-19 was being diagnosed with PCR from nasal and oropharyngeal swabs with a duration of 1–4 days where the sample was tested both locally in the hospital lab and corroborated with national lab testing to reduce risk of unequivocacy. Groups of patients were divided into either not swabbed (due to being asymptomatic) or swabbed due to presence of documented symptoms which yielded either negative or positive results.

## Statistical analysis

All the data were recorded, anonymised and verified by four members of the study group for their accuracy. The data were processed using Microsoft Excel (Microsoft, Washington, USA). Shapiro-Wilk test indicated a normal distribution for age and days to discharge from hospital; hence, the mean (±SD; 95% CI) were calculated for both. ASA did not follow normality and was analysed using median (±median absolute deviation (MAD)) and IQR. Both prevalence ratio (PR) or risk ratio (RR) and odds ratio (OR) were calculated as well as a Fisher's exact test for statistical significance for categorical data, defined as p≤0.05. Percentages and CIs were rounded off to one decimal place.

## The collaborative

The COVid Emergency-Related Trauma and orthopaedics (COVERT) Collaborative was founded at Imperial College Healthcare NHS Trust. It is currently a member of the COVID-19 Research Group and it has been endorsed by the Royal College of Surgeons of England and Imperial College Healthcare NHS Trust.

## Patient and public involvement

Patients and the public were not involved in the study design, recruitment or conduct.

## RESULTS

A total of 4840 clinical encounters took place between the study periods. A total of 172 spinal operations and presentations were excluded from the final analysis. Table 2 outlined demographic data. During the COVID-19 period, there was a 34% reduction in acute orthopaedic trauma referrals compared with 2019 (1792 down to 1183 referrals), and 29.5% less surgical interventions (993 down to 700 operations). Figures 1–3 have categorised these clinical encounters into types and mechanisms of injury for both acute referrals and operative cases between both years, respectively.

## COVID-19 status

COVID-19 status for both acute referrals and operative casemix including results in all mortalities was demonstrated in figure 4. Mortalities with positive swab results were confirmed prior to the event of death within 6 weeks postpresentation. Approximately three-quarters were not tested and a fifth had negative results. 3.2% of the acute referrals resulted in deaths and 28.9% of those mortalities (ie, 0.9% of the entire cohort) tested positive for COVID-19. Furthermore, postoperative mortalities represented 4.9% of the entire operative casemix, and 32.4% of those mortalities (ie, 1.6% of the entire cohort) had a confirmed positive COVID-19 diagnosis prior to their death.

## Risk (or prevalence) ratio and OR

Table 3 outlined the RR (or PR) and OR alongside their 95% CIs and statistical significance. The RR is synonymous with the PR. Only those factors that were statistically significant within the acute referrals and operative caseloads were included. There were trends demonstrating increase in

**Table 2** Demographic data of pre-COVID-19 and post-COVID-19

| | | Pre-COVID-19 (2019) | | COVID-19 (2020) | |
|---|---|---|---|---|---|
| Acute referrals | **Total** | **1792** | | **1183** | |
| | Male | 935 | 52.2% | 560 | 47.3% |
| | Female | 857 | 47.8% | 623 | 52.7% |
| | Mean age±SD (95% CI) | 52.2±27.9 (50.9 to 53.5) | | 55.8±27.9 (54.3 to 57.4) | |
| Operative cases | **Total** | **993** | | **700** | |
| | Male | 499 | 50.3% | 320 | 45.7% |
| | Female | 494 | 49.7% | 380 | 54.3% |
| | Mean age±SD (95% CI) | 51.7±28.1 (50 to 53.5) | | 57.7±26.7 (55.7 to 59.6) | |
| | Median ASA±MAD (IQR) | 2±1 (2) | | 2±1 (2) | |

ASA, American Society of Anesthesiologists; MAD, median absolute deviation.

mortality rates, use of anaesthetic AGPs, consultant-led operations, hip fracture surgery and falls; but a decrease in other lower limb operations, open reduction and internal fixation (ORIF), removal of metalwork and foreign bodies (FB), road traffic accidents/road traffic collision (RTA/RTC), sporting injuries and infection.

## Mortality

Table 3 indicated that the 6-week mortality rate more than doubled significantly (p<0.01) for both cohorts for RR (RR=2.19–2.50) and OR (OR=2.25–2.55) during the COVID-19 period. COVID-19-related complications were still responsible for increasing the odds of mortality by 20–22 times within all mortalities from both acute referrals and operative cases (as compared with non-COVID-19 causes for all mortality in the year 2019). Table 4 confirmed that the mean age of mortalities across the board was in the elderly patient population with a high median ASA grade. Males were consistently in the minority, while neck of femur (NOF) fracture was the modal diagnosis due to falls and persistently in the majority, followed by lower limb injuries (figures 5 and 6). At least 82% of operations were related to neck of femur fractures in which half of all operations during the COVID-19 period involved anaesthetic AGPs. Whereas the mortalities from pre-COVID-19 operations did not have consultant-led (as primary surgeon) surgery, that increased to three-fifths of all operations conducted during the COVID-19 period (figure 7).

## Subgroup analysis for NOF fracture

A subgroup analysis of hip fractures was conducted due to its recognised risk of mortality within orthopaedic trauma. Those who were operated on in 2020, 20.2% tested positive for COVID-19, 47.3% tested negative and

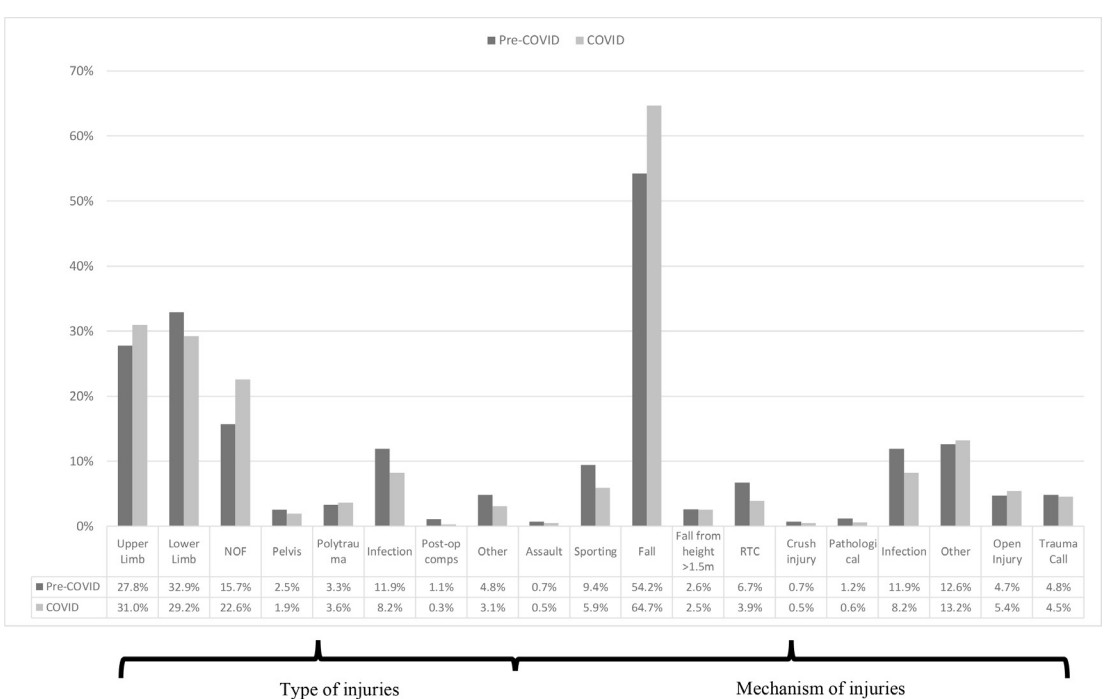

**Figure 1** Types and mechanisms of injuries for acute referrals. Comps, complications; NOF, neck of femur fracture; RTC, road traffic collision.

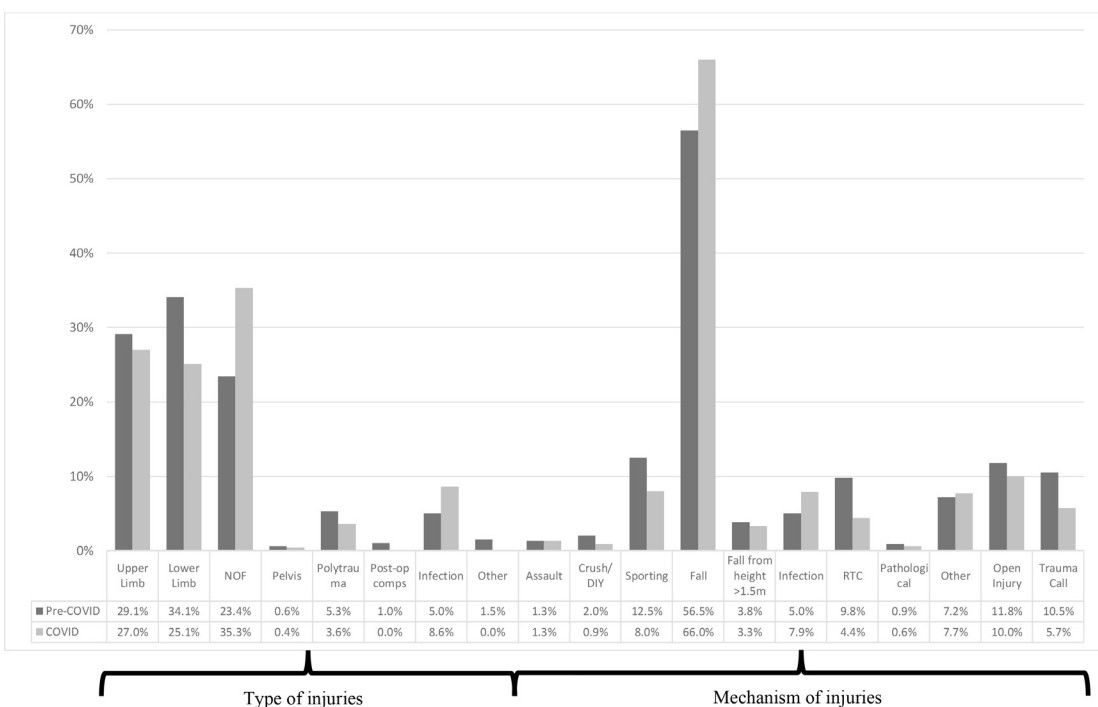

**Figure 2** Types and mechanisms of injuries for operative cases. Comps, complications; DIY, do-it-yourself; NOF, neck of femur fracture; RTC, road traffic collision.

the remaining 32.4% were not tested due to being asymptomatic. Furthermore, 82.3% of all mortalities in 2020 sustained a neck of femur fracture in which only 35.7% of this cohort had a positive swab result, 21.4% with negative swab results and the remaining 42.9% were not swabbed due to being asymptomatic. There was no statistical difference in the OR and RR between both years for

mortality rate in NOF fractures (table 3). The absolute numbers did not change much, but because of a drop in other presentations, the relative percentage of NOF fractures markedly rose. Hence, the mortality expressed as a percentage of cases is notably higher for all operations, and not necessarily if stripped down to hip fractures alone.

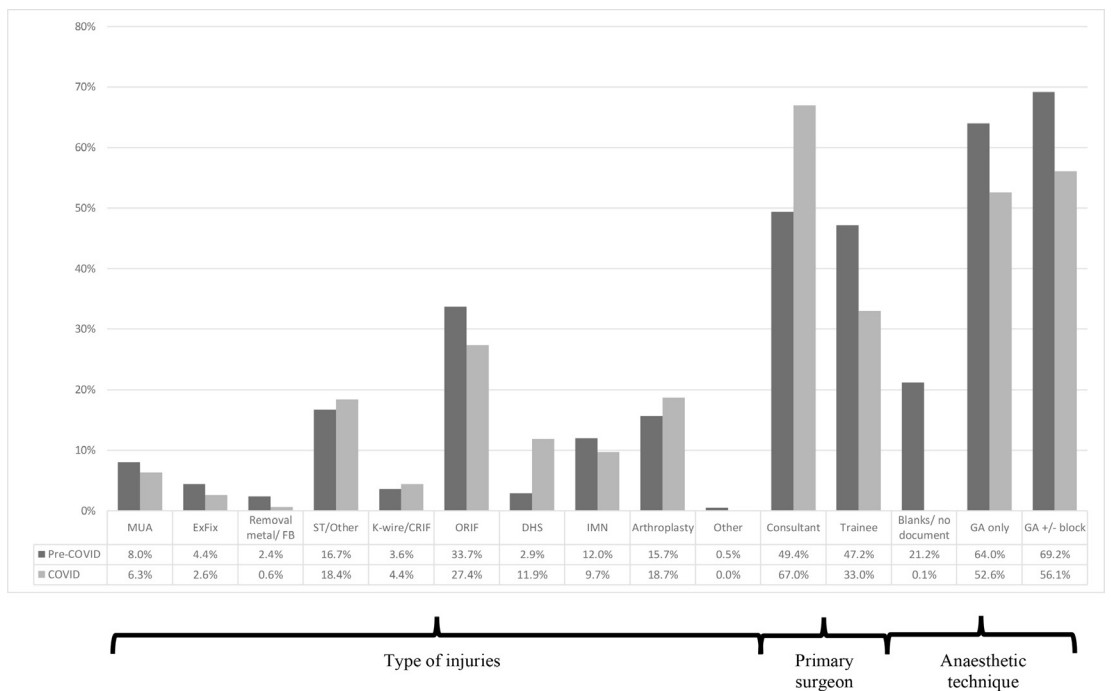

**Figure 3** Operative and anaesthetic techniques compared between pre-COVID-19 and post-COVID-19. CRIF, closed reduction and internal fixation; DHS, dynamic hip screw; Ex-fix; external fixation; FB, foreign body; GA, general anaesthetic; IMN, intramedullary nailing; MUA, manipulation under anaesthesia; ORIF, opeen reduction and internal fixation; ST, soft tissue.

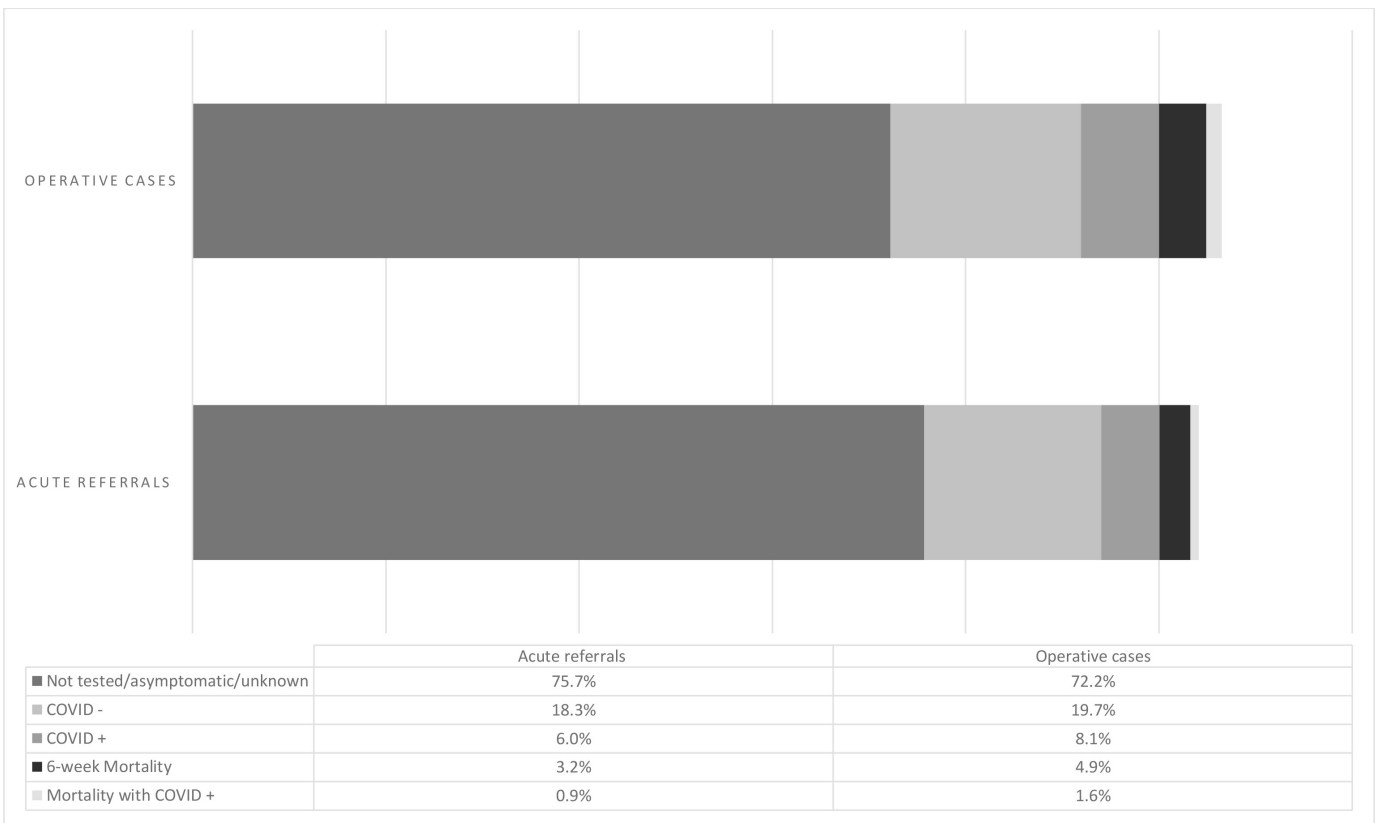

| | Acute referrals | Operative cases |
|---|---|---|
| ■ Not tested/asymptomatic/unknown | 75.7% | 72.2% |
| ■ COVID - | 18.3% | 19.7% |
| ■ COVID + | 6.0% | 8.1% |
| ■ 6-week Mortality | 3.2% | 4.9% |
| ■ Mortality with COVID + | 0.9% | 1.6% |

**Figure 4** COVID-19 status for acute referrals and operative cases as a measure of proportions.

### Preoperative and postoperative morbidity

Taking into account that COVID-19 was a perioperative complication since patients may have been symptomatic with COVID-19 manifestations preoperatively but only had the swab results return with a positive finding either preoperatively or postoperatively; the most common postoperative complication in the COVID-19 period was a hospital-acquired pneumonia but with negative COVID-19 swab results or the decision not to test at all. The second most common postoperative complication in the year 2020 was extra-pulmonary sepsis (online supplemental appendix 1). The proportion of postoperative complications had significantly increased when including or excluding COVID-19 as a perioperative or postoperative complication in 2020 (0.70% vs 2.57–4.14%; p=0.003) with varying OR (3.72–23.4; p<0.01) and RR (3.65–32.6; p<0.01) (table 3). Online supplemental appendices 2 and 3 focused on the total number and nature of comorbidities within the mortality groups. Multiple contingency $\chi^2$ test was insignificant for both number of comorbidities and individual comorbidities between both years, except for cardiovascular and cerebrovascular disease in acute referral cohort.

### Survival probability

Six-week Kaplan-Meier survival probability analysis for mortalities between both years is plotted in figures 8 and 9. There were similar patterns of survival probability between both cohorts (ie, 2019 vs 2020 cumulative). However, the lowest survival probability and the shortest timeframe were observed in the confirmed COVID-19 positive cohorts (figure 8). Eight (72.7%) patients had femoral trauma, most being neck of femur fractures, distal femur fracture and a dislocated hip hemiarthroplasty post-fracture. Unexpectedly, there was a reversal of trends observed for the 6-week Kaplan-Meier survival analysis once admitted and operated in figure 9. Mortalities within the pre-COVID-19 period had the lowest survival probability compared with the post-COVID-19 cohort. The COVID-19 positive mortalities were observed to have the highest survival probability of 11 days prior to converging with those mortalities without COVID-19 symptoms.

### DISCUSSION
### Comment on alternative hypothesis

There was a significant difference between pre-COVID-19 and post-COVID-19 periods at its peak. The alternative hypothesis was not rejected with respect to prevalence of (1) acute orthopaedic trauma referrals (reduced by 34%), (2) surgical interventions (reduced by 29.5%), (3) anaesthetic AGPs, (4) 6-week mortality rates (more than doubled in the COVID-19 period) and (5) survival probability between pre-COVID-19 and post-COVID-19 eras.

### Corroboration of our results with current literature

The 34% reduction in acute trauma referrals is in keeping with previous single-centre studies performed in the UK with results ranging between 26% and 59%.[10–13 15 16] As

**Table 3** Risk (or prevalence) ratio and OR for acute referrals and operative caseloads. Comparisons are made between COVID-19 period against the pre-COVID-19 period. Value >1 indicated greater odds or risk during the COVID-19 period

| | | Acute referrals | | | Operative caseload | | |
|---|---|---|---|---|---|---|---|
| | | RR | OR | P value | RR | OR | P value |
| Morbidity and mortality | Mortality | 2.50 | 2.55 | 0.0005 | 2.19 | 2.25 | 0.004 |
| | Mortality due to COVID-19-related complications versus non-COVID-19 causes | 14.2 | 19.7 | 0.004 | 15.1 | 22.0 | 0.004 |
| | Perioperative/postoperative complications including COVID-19 | | | | 5.88 | 6.09 | 0.00001 |
| | Perioperative/postoperative complications excluding COVID-19 | | | | 3.65 | 3.72 | 0.003 |
| | Perioperative/postoperative COVID-19 positive testing | | | | 32.6 | 23.4 | 0.0009 |
| Anaesthetic technique | General anaesthetic only | | | | 1.22 | 1.61 | 0.00001 |
| | General anaesthetic±block | | | | 1.23 | 1.75 | 0.00001 |
| Consultant involvement | Consultant-led operation | | | | 1.36 | 2.08 | 0.00001 |
| Operation technique | Open reduction+internal fixation | | | | 0.81 | 0.74 | 0.007 |
| | Dynamic hip screw | | | | 2.02 | 2.11 | 0.00001 |
| | Removal of metal/foreign body | | | | 0.24 | 0.23 | 0.003 |
| Mechanism of injury | Road traffic accident | 0.58 | 0.56 | 0.001 | 0.45 | 0.43 | 0.00001 |
| | Fall (<1.5 m) | 1.19 | 1.54 | 0.00001 | 1.17 | 1.49 | 0.0001 |
| | Sporting injury | 0.63 | 0.60 | 0.0005 | 0.64 | 0.61 | 0.003 |
| | Infection | 0.69 | 0.66 | 0.001 | 1.70 | 1.77 | 0.005 |
| | Trauma call | | | | 0.55 | 0.52 | 0.0005 |
| Type of injury | Neck of femur (NOF) fracture | 1.44 | 1.57 | 0.00001 | 1.51 | 1.79 | 0.00001 |
| | Lower limb (excl. NOF) | 0.89 | 0.84 | 0.04 | 0.74 | 0.65 | 0.0001 |
| Gender | Male | 0.91 | 0.82 | 0.01 | | | |

OR, odds ratio; RR, risk ratio.

described in these previous studies, we would attribute the overall reduction of trauma workload to be due to reduction in travel and outdoor activities during the national lockdown. MacDonald et al[17] described a similar effect in their multicentre study with a reduction of operative workload by 26.5% compared with 29.5% in our study. Sites recruited for this study confirmed that they continued to operate at their own facilities during the data collection period whereas some later used alternative and external facilities including private hospitals through NHS England pathways (as mentioned by Dayananda et al[18]), which may have impacted nosocomial rates of COVID-19, morbidity and mortality. However, this would be difficult to assess since it would also depend on the diversions of the ambulance services to 'clean' versus 'contaminated' hospital sites.

## Changes in trends during the peak of COVID-19

With respect to the operative caseload (table 3), patients had half (OR=0.52, p<0.001) the odds of presenting as a trauma call. This was due to the ORs of road traffic accidents, sporting injuries, infection and lower limb injuries were significantly less (by 34%–44%; OR=0.56–0.66, p<0.01) during the COVID-19 period. Conversely, there was a significant rise in the odds of neck of femur fractures, falls, the use of anaesthetic AGP and consultant-led operations; a finding also reflected by Arafa et al.[19]

Although the expectation was to minimise the use of aerosolising-generating anaesthetic procedures, there was in fact an increased prevalence of using general anaesthesia±block up to an odds of 75%, in order to create a 'closed circuit' for the airways. As the anaesthetic methods was not well documented in the pre-COVID-19 era in a fifth (21.3%) of cases, this skewed the data as it may have been difficult to extract that data from 2019. The odds of a consultant-led operation doubled (OR=2.08) during the COVID-19 period as a consequence of all elective operations being suspended, hence more consultants were relocated to trauma theatre and increased pressure within the theatre environments led to consultant-delivered, rather than consultant-led care. With respect to surgical procedures, there was a significant reduction in PR of open reduction and internal fixation by a fifth

**Table 4** Patient demographics, date of injuries and time to mortality

| | Acute referrals | | Operative casemix | |
|---|---|---|---|---|
| | **2019 (n=23)** | **2020 (n=38)** | **2019 (n=22)** | **2020 (n=34)** |
| Mortality | 1.3% | 3.2% | 2.2% | 4.9% |
| Mortality with COVID-19 positive PCR result | | 0.9% (total) 28.9% (mortality cohort) | | 1.6% (total) 32.4% (mortality cohort) |
| Post-op morbidity | | | 0.7% | 4.1% |
| Age (years; mean±SD; 95% CI) | 80.2±16.4 (73.2 to 87.2) | 77±23 (67 to 88) | 83.9±12.2 (78.7 to 89.1) | 84.0±13.5 (79.4 to 88.5) |
| Male | 9 39% | 16 42% | 8 36% | 15 44% |
| ASA (median±MAD; IQR) | | | 3±0 (1) | 3±0 (0) |
| Date of injury (mean days±SD; 95% CI) | 6/4±11 (1/4 to 10/4) | 31/3±12 (26/3 to 5/4) | 6/4±12 (1/4 to 11/4) | 30/3±14 (25/3 to 4/4) |
| Time from admission to mortality (mean days±SD; 95% CI) | 10.3±7.5 (7.1 to 13.5) | 11±10 (7 to 15) | 14.3±10.4 (9.8 to 18.7) | 13.8±10.4 (10.2 to 17.3) |

ASA, American Society of Anesthesiologists; MAD, median absolute deviation.

(PR=0.81) and removal of metalwork and foreign bodies by three-quarters (PR=0.24), while there was a doubling (PR=2.02) in dynamic hip screw fixation in the COVID-19 era.

## Mortality and morbidity
### Mortality during COVID-19 timeframe
Comparatively, the COVIDSurg Collaborative observed a mortality rate of 28.8% (p<0.0001) of orthopaedic patients who underwent surgery (both elective and trauma) within the first quarter of the year.[20] The increased mortality during the pandemic is partly due to selection of cases that required surgical intervention. The decrease in acute

referrals and operations indicated a higher threshold for treatment (due to a redistribution of hospital resources during the pandemic). However, no such case was denied surgery but in the worst-case scenario, patients were offered postponed treatment which is acceptable practice (ie, within 2 weeks).

### Role of morbidity in mortality during COVID-19
Results from figures 5–7 and online supplemental appendices 1–3 were corroborated with the COVIDSurg publication[20] which confirmed a significant association of mortality with myocardial infarction and congestive heart failure. However, hypertension and stroke/transient

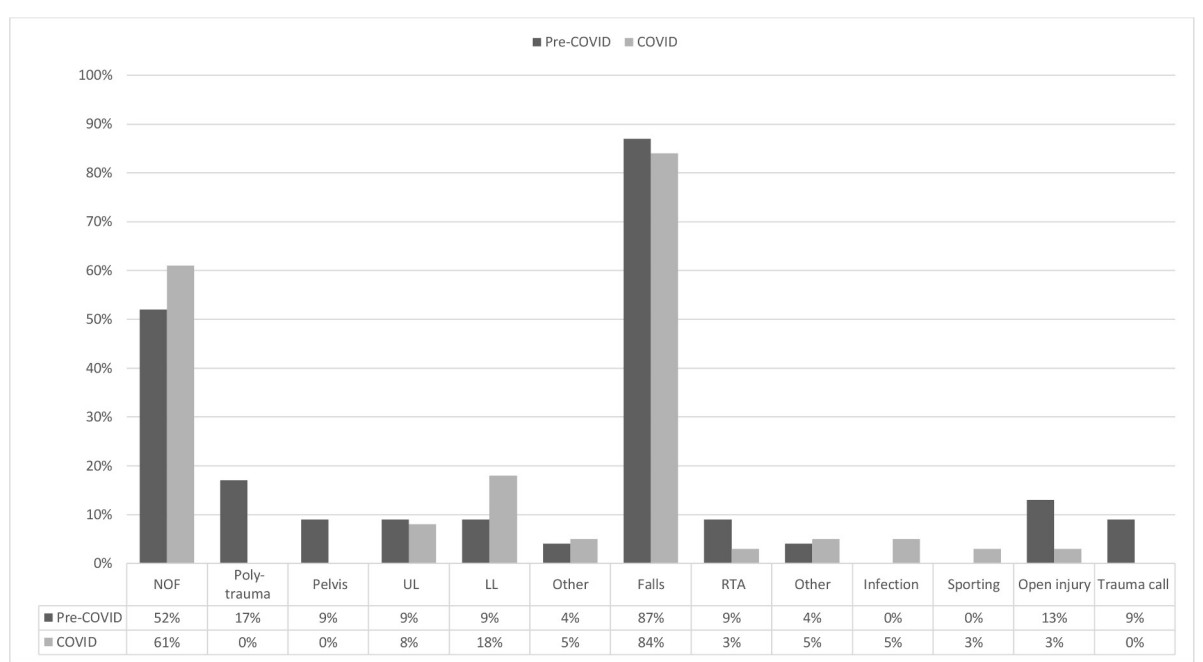

**Figure 5** Types and mechanism of injury for mortalities in acute referral cohorts. LL, lower limb; NOF, neck of femur fracture; RTA, road traffic accidents; UL, upper limb.

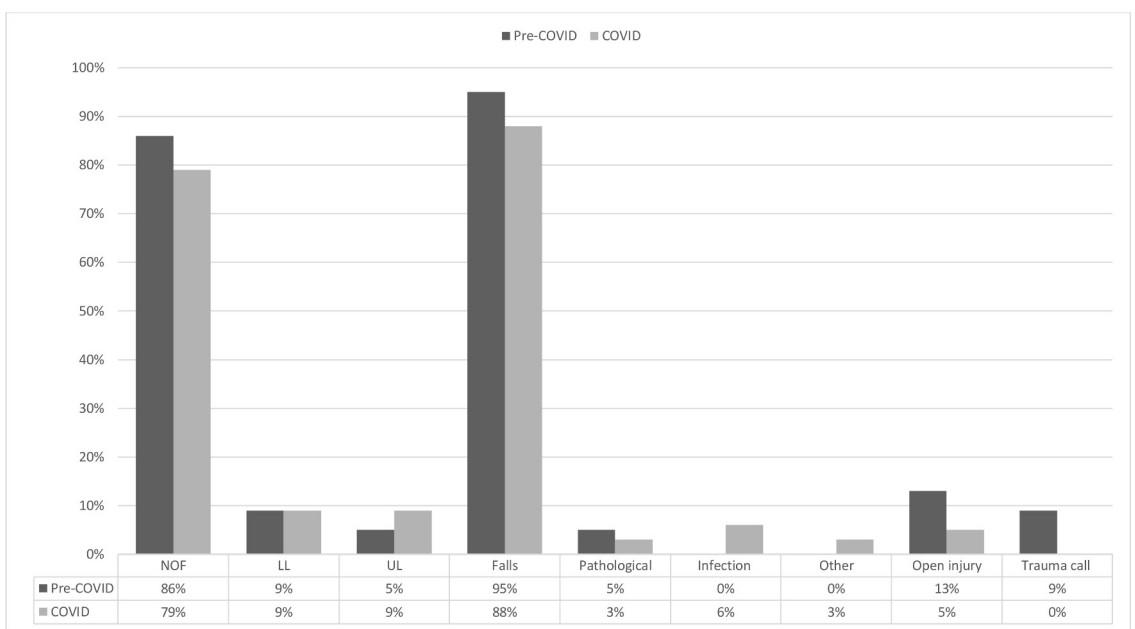

**Figure 6** Types and mechanism of injury for mortalities in operative casemix cohorts. LL, lower limb; NOF, neck of femur fracture; UL, upper limb.

ischaemic attacks were not significantly associated. In our study, all cardiovascular diseases (including peripheral vascular, arrhythmias, hypertension, heart failure, myocardial infarction and acute coronary syndromes) were combined with cerebrovascular diseases (consisting of strokes and transient ischaemic attacks). Unlike their study, our study did not find a significant association with chronic kidney disease, chronic obstructive disease (which included asthma) and dementia in all mortalities during the 2020 timeframe regardless of the COVID-19 status. The differences may stem from that their study looked at the comparison of mortality rates within the

same cohort during the COVID-19 era, whereas this study is sub-analysing the entire mortality cohort on its own to observe for specific associations and risks.

### Survival probability between both years

As expected, reduced survival probability reflected the most vulnerable patient profiles, usually with multiple preoperative and postoperative comorbidities (online supplemental appendices 1–3). A reason for a transient increase, and unexpected reversal, in 6-week survival probability in the operative COVID-19 cohort (figures 8 and 9) may be explained by dedicated wards

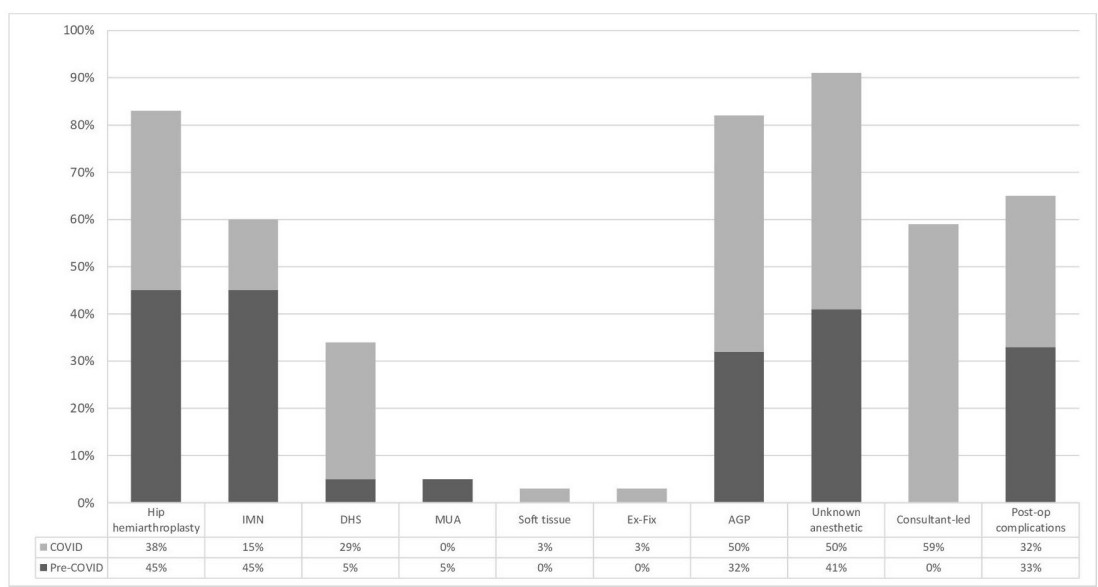

**Figure 7** Surgical and anaesthetic techniques used in mortalities as a means of proportions. AGP, aerosolising-generating procedures; DHS, dynamic hip screw; Ex-fix; external fixation; IMN, intramedullary nailing; MUA, manipulation under anaesthesia.

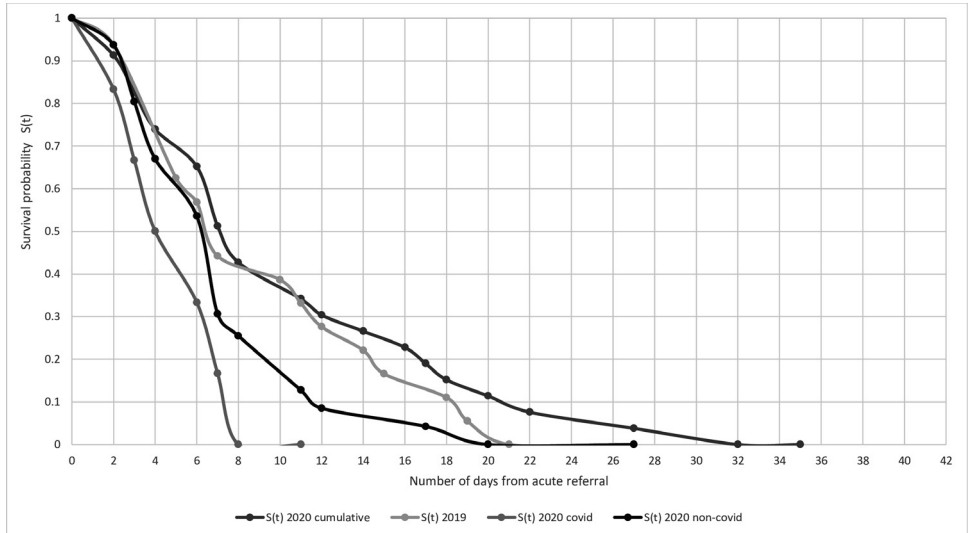

**Figure 8** Six-week Kaplan-Meier survival probability analysis for mortalities between pre-COVID-19 and post-COVID-19 for acutely referred from the emergency department.

being ring-fenced to host confirmed COVID-19 positive patients with a heightened care of nursing, medical cover and personal protective equipment. Prior to the onset of a possible vaccination, symptomatic management and shielding were the mainstay treatments for COVID-19 positive patients. None of these patients were stepped up to the intensive treatment unit due to being categorised as high-risk stratification for mortality based on age and extent of comorbidities.

### Justification of conducting this study

As lockdown measures in the UK and globally eases and the incidence of trauma returns to pre-lockdown trends, it is imperative that we understand the true increased risk of mortality in acute trauma during the COVID-19 era. A recent publication by Kader et al[21] has suggested that the

rate of mortality from COVID-19 for elective orthopaedic patients is low; yet this is the first British multicentre study to quantify mortality risk for trauma patients. Trauma procedures due to the nature of the injuries are necessary and time-critical, and nobody can afford to postpone trauma care even during a global pandemic.[22]

Furthermore, the Corona Hands Collaborative[23] published that upper limb trauma patients had SARS-CoV-2 complication rate of 0.18% (n=2) with 0.09% (n=1) overall mortality at the peak of the first wave in April 2020. However, their collaborative looked into a shorter postoperative period (30 vs 42 days) but they agreed that patients who had been hospitalised for a prolonged period before their surgery were at increased risk of both COVID-19-related and postoperative complications. Most of their

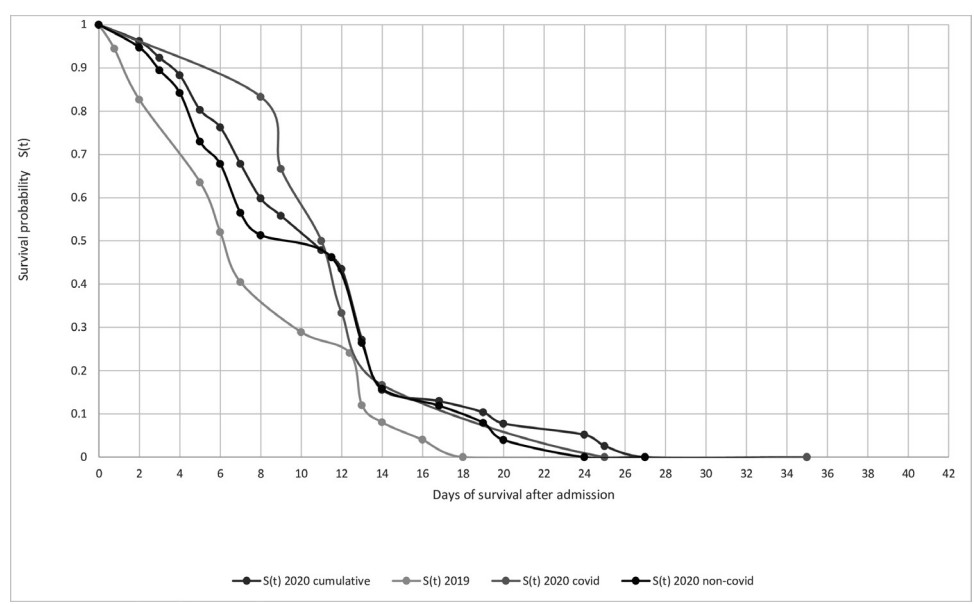

**Figure 9** Six-week Kaplan-Meier survival probability analysis for mortalities between pre-COVID-19 and post-COVID-19 for those undergoing surgery.

patient cohort, who were both younger and fitter than our cohorts, would be classified as the 'walking wounded' and could usually be managed as day-case procedures.

Although the trends in mechanisms of injury in our study were reflective of those within a US multicentre study, there was an opposing trend in the number medical/surgical procedures.[24] That could be due to their study encompassing level 1 trauma centres with a mean younger patient population. However, we do agree that with time and from experiential learning, hospitals improved their coping strategies with the pandemic and enhanced patient safety by enforcing personal protection equipment, hosting dedicated theatres for COVID-19-positive patients, separating sites as clean and contaminated, ringfencing COVID-19-positive patients to dedicated wards and promoting routine COVID-19 PCR swabs for all admissions and preoperative checklists.

With an overall mortality risk in 2020 doubled that of 2019, clinicians need to counsel patients presenting with acute orthopaedic trauma of the increased risk in the COVID-19 era, especially for those identified as increased risk stratification with multiple underlying comorbidities, elderly and frailty. With the ongoing risk of a subsequent wave and resurgence of COVID-19 cases on top of the inevitable winter pressures, this data is of critical importance in the risk management, decision-making and policymaking of trauma patients both in the UK and across the globe.

### NOF fractures

Since the aetiology of neck of femur fracture is often low energy falls in the home environment, it is not unexpected to observe a consistency of neck of femur fractures in the elderly and the vulnerable during lockdown as seen in figures 1 and 2. Those with NOF fractures remain at greatest risk of mortality and there have been further studies evaluating the risk of COVID-19 on this inherently high-risk cohort.[25–28] COVID-19 itself has been identified as an independent risk factor in increasing mortality in neck of femur fractures.[29 30]

The increased mortality reflect the increased proportion of patients with NOF fractures that have a higher baseline mortality which has been echoed by the Scottish IMPACT-Restart study.[28] There are several justifications such as reduced help, lack of assistance and staff shortages due to the effect of the national lockdown which required elderly patients to be more independent, unsupervised and at higher risk of falling. Nevertheless, it should be considered that odds of falls may have increased due to prodromal symptoms and clinical manifestations of COVID-19.

If these 'at risk' patients were symptomatic with the virus, then aggressive preoperative optimisation would occur. Since 91% (n=10) of COVID-19 positive patients had sustained a NOF fracture, the National Hip Fracture Database (NHFD) best practice tariff of operating within an ideal 36-hour window set by the Royal College of Physicians was suspended until the patient was stabilised. All

hip fracture patients in this cohort were operated on and had dedicated orthogeriatric input commencing from hospital admission. Hence the early perioperative period and surgery encompassed within the 10-day period post-admission. Moreover, NOF fractures are recognised as a pre-terminal illness and are known to carry a high risk of mortality in the first month which is trebled in the first year after the injury.[31]

### Strengths and weaknesses of the study and in relation to other studies

This was the first representative observational multicentre study of the UK looking into the impact of COVID-19 pandemic on general trauma and orthopaedic surgical practice. Studies thus far have only shed light on local scales, cross-speciality, reflecting a fraction of our study population or contain 30-day mortality at most.[10–13 20 30 32 33] Weaknesses included loss of data points which have been accounted for in the tables (ie, labelled as unknown). However, this did not affect the final analysis of data points (table 1). Operations conducted outside the specific study periods will not account for all those operations required such as for NOF fractures. It does not suggest that the number of NOF fractures not accounted for have been managed conservatively (as discovered by Cherevu *et al*[34]), since some NOF fractures may have breached time to surgery due to medical reasons or being influenced by international guidelines.[35]

### Limitations and future research

It is vital to continue exploring the impact of the pandemic on a larger scale. Ideally, more secondary care providers consisting of district general hospitals and major trauma centres will submit data. The diagnosis of COVID-19 was dependent on positive PCR swabs for this study rather than non-specific changes seen on chest CT or plain radiographs. This does not account for false negatives with clinical respiratory symptomatology or true positives in those asymptomatic. Nevertheless, this issue with data has been speculated on in another national study.[23] Data ought to be submitted during the peak of the pandemic as well as at various time intervals as the lockdown measures ease resulting in more freedom of movement while also accounting for the continued risk of subsequent waves and national lockdowns.[36] Further studies will also require to compare the impact of the pandemic on the specialty in the UK compared with other countries on other continents.

### CONCLUSION

This was the first, longest and largest British multicentre representation of the impact of COVID-19 pandemic on acute orthopaedic trauma referrals and mortality between mid-March and end-April 2020, representing the peak of the first wave during the lockdown. The mortality rate for acute referrals, as well as those undergoing operative intervention, more than doubled in odds when

compared with the same time interval 1 year ago. The majority of mortalities consisted of the elderly with neck of femur fractures and cardiovascular and/or cerebrovascular diseases. This study will aid clinicians in counselling trauma patients of the increased risk of mortality during the era of COVID-19 and also aid in both healthcare infrastructure, resource allocation, decision-making and policymaking as we continue to battle with the pandemic.

**Collaborators** COVERT Collaborative (Individual names have been added after references as per your request including their extent of contribution and contact details) Thomas Ashdown; Ramla Ali; William Harland; Charles Hallett; Chinmay Gupte; Dominic Spicer; Robin Strachan; Dinesh Nathwani; Rajarshi Bhattacharya; Catrin Morgan; Katharine Bennett-Brown; Henry Simon; Rupen Dattani; Adham Mousa; Karen Vejsbjerg; Michael Zhang; Vikas Kumar; Hannah Emerson; Andrew Oliver; Callum Smith; Andreas Hinsche; Markus L Sagmeister; Gihan Jayasinghe; Marissa Lewis; Michael Avery; Tariq Aboelmagd; Shireen Ibish; Ruben Thumbadoo; Asanka Wijendra; Joseph Dixon; Vatsal Gupta; Laura Clifton; Suhib Taher; Nicholas Jia Jie Wei; Emma Reay; Ashish Khurana; Kathryn Rooney; Nakanda Kakaire; Oluwawyemisi Adesina.

**Contributors** KS is the guarantor and has overall responsbility of the content. We have read the ICJME guidelines attentively and have outlined the collaborative contributors below who have all satisfied the criteria to be recognised as a collaborative co-author if this study is published. The core committee of the collaborative consists of KS, AA, CP and KMS who all conceptualised the study, led the planning/investigation/methodology/design, supervised the collaborative contributors' roles, as well as the initial and final version of the manuscript. Additionally, KS (primary author) was also leading data curation, project administration and resource allocation with KMS. KS was also leading on validation. KS, CP and AA led the data analysis and the reporting of the results. All other collaborative members, from the seven centres and outside the core committee, were involved in data curation, formal analysis and in resource allocation internally under consultant supervision (ie, those with FRCS). Individual contributions from the core committee and every collaborative member has been outlined below.

**Funding** The authors have not declared a specific grant for this research from any funding agency in the public, commercial or not-for-profit sectors.

**Competing interests** None declared.

**Patient consent for publication** Not applicable.

**Ethics approval** This study involves human participants but an Ethics Committee(s) or Institutional Board(s) exempted this study. All data points were used for routine auditing purposes to reflect departmental activity and service provision without altering clinical care pathways. Each centre contributing data to this study registered their interests with local authority and the auditing or clinical governance departments. No informed consent was required as there were no identifiable data. All data were anonymised at the time of collection and submission. Each patient was assigned a unique identification number which was cross-referenced with the patients' individual hospital identification or medical record numbers. This cross-referenced list remained internally within the hospital trust computer server handled by the contributing team from each trust. The data were transferred and stored using the NHS.net email server which has been approved for transfer of patient data. Data protection compliance was abided by at all times. The lead centre was Imperial College Healthcare NHS Trust where this study was first approved as a clinical audit prior to expanding onto a national scale. All centres gave permission for the use of their data. This study was assessed using the UKRI/MRC/NHS Health Research Authority Ethics Decision Tool and was considered an 'audit/not research'; and therefore it was not subject to further ethical review by the NHS Research Ethics Committee (NHS REC).

**Provenance and peer review** Not commissioned; externally peer reviewed.

**Data availability statement** Data are available upon reasonable request. All data relevant to the study are included in the article or uploaded as supplementary information. Underlying data, code and supporting documentation may be made available as a redacted version to interested parties, subject to the completion of a protocol and signing of a Data Transfer Agreement.

**ORCID iD**
Kapil Sugand http://orcid.org/0000-0003-3185-954X

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
