## [Reviewer comments · BMJ Open]

ARTICLE DETAILS

TITLE (PROVISIONAL)	The impact of COVID-19 on acute trauma and orthopaedic referrals and surgery in the UK during the first wave of the pandemic: a multicentre observational study from the COVID-Emergency Related Trauma and orthopaedics (COVERT) Collaborative
AUTHORS	Sugand, Kapil; Aframian, Arash; Park, Chang; Collaborative, COVERT; Sarraf, Khaled

VERSION 1 – REVIEW

REVIEWER	Prada, Carlos McMaster University, Orthopaedics
REVIEW RETURNED	28-Jul-2021

GENERAL COMMENTS	Main comments: 1. Relevance: I have to start saying that I acknowledge the effort behind this initiative. However, the results are not a surprise and do not add significant information to the current discussion and what has been published recently on this topic. Despite being a big study with robust collected data the novelty of this manuscript is minor.2. Results: In more than one opportunity in the results section the authors provide explanations to certain factors/outcomes. I suggest to remove those paragraphs and address those aspects in the discussion. The result section should just show a summary of the collected data in a friendly way to the readers without judgment or appraisal.3. Discussion:A. The authors suggest that the increase in the incidence of neck femoral fractures could have been explained by prodromal symptoms of COVID-19, without clarifying what was the percentage of patients with these fractures that were COVID-19 (+) to weight the relevance of this argument. In addition, it seems more likely that given that elderly people sometimes need help at their home and could have lacked of the assistance they used to have because of the national lock-down, so the need to perform tasks by themselves could have put them at a higher risk.B. What are the causes behind the increase in the overall mortality in the authors opinion? It is unclear to me what factors they attribute to be the cause(s) behind this difference, specially in the COVID-19 (-) population. This is a key aspect of the manuscript and definitely needs to be developed more exhaustively. Minor comments:
---

	- “Golden peak” concept: No references are supporting the decision of declaring this 6-week period as the peak. Epidemiological COVID-19 data from those weeks should be added and referenced to give the readers perspective of the magnitude of the first wave in the included territories to help with external validity of the study. - Statistics: I suggest to check that the ASA grade had a normal distribution. Commonly, this variable is non-normally distributed. If that’s the case, 95% CI are not preferred for the table showing ASA grade. 95% CI showing 2-2 is not informative so, as this variable is probably skewed, median and interquartile ranges might be needed instead. Thank you again for submitting your work.
--	--

REVIEWER	Ohm, Eyvind Norwegian Institute of Public Health, Department of Health and Inequality
REVIEW RETURNED	10-Aug-2021

GENERAL COMMENTS	Thank you for the opportunity to review this interesting manuscript, which significantly adds to our knowledge about the impact of COVID-19 on trauma care and orthopaedic workload. Specifically, the authors document a reduction in trauma referrals and surgical interventions during the early phase of the pandemic, combined with an increase in mortality for this patient population. Major strengths include the assessment of mortality, and details about COVID status of trauma patients. However, I do have some comments about the manuscript. The Results section needs some work, and some text passages seem oddly placed (see below for details). Abstract: under “Setting” it is claimed that hospitals from seven cities were recruited. However, from the description of the contributing hospitals (pages 3-9), I count seven hospitals from six cities (two hospitals are located in London). Further, I would have liked to see a list of these hospitals in the main text (e.g., in Methods), since it is claimed (without documentation) that this study is representative of the whole UK. Introduction: It seems redundant to describe the outcomes in the Introduction, since this information is repeated in Methods (where such details belong). Methods:  • The date given for the end of the study period in 2020 (page 16) is wrong (should be Tuesday the 28th of April, not the 31st) • Inclusion/exclusion criteria: Elective procedures were excluded, yet “patients listed for an operation prior to time period of data collection were included” – this is confusing to me. The latter operations seem elective in nature, since these traumas could be postponed until a later date. • COVID status: are the three groups swabbed mutually exclusive? Tests on those swabbed due to presence of documented symptoms (group 2) could result in both negative (group 3) or positive (group 4) results. Or? Results:
--

- The number of tables/figures must surely exceed the maximum allowed for this journal? Some figures are not very informative (e.g., Figure 6) and could be summarized in the text.
- I would have liked to see a sentence or two describing the main results from Table 2, documenting the overall change in referrals/surgical interventions (as done in the Abstract). Likewise, the results from the main descriptive analyses shown in Figures 1 and 2 could briefly be summarized to get an overall picture of changes witnessed during the pandemic.
- Figure 1: why is “infection” both an injury type and mechanism? Seems redundant to include twice in the same figure.
- In Figure 1, the bar to the left (in dark) refers to the pre-covid period while the bar to the right (in grey) refers to the covid period. In Figure 2 this is the other way around, appearing backwards in time.
- COVID status: information about testing (first sentence on page 21) belongs in Methods.
- I found Figure 4 (“COVID status of all mortalities”), and the text on page 25, difficult to understand. 6-week mortality is straightforward, but what does 6% and 8.1% for COVID positive during the COVID period mean? Is this the percentage of dead patients with a positive COVID result, or the percentage of patients with COVID who died? Or something else? Presentation of these percentages was confusing to me.
- Table 3 (and the text introducing this table on page 23): ratios here refer to comparisons between pre-COVID vs. COVID periods, but this is not made explicit in the caption/title.
- Some of the text on pages 23-24 sounds like discussion (“...perhaps to create a closed circuit...”, “...this could have skewed the data...”, “...as a consequence of all elective operations being suspended...”, “This could be due to..”). I would rephrase this and/or move these speculations to the Discussion.
- Figure 7a/b: I found these figures difficult to read. Bar charts may be more intuitive to get the message across. Further, much of the text regarding comorbidities focuses on comparing results from other studies (page 27) and belongs in the Discussion.
- Survival analyses: how can mortality rates for operative cases double during the COVID period (Table 3), but survival probability be lowest pre-COVID?
- The last two paragraphs of Results (pages 29-30) sound more like background/discussion.

Discussion:

- Mortality: “...and a third (29-32%) of those deaths had a positive COVID-19 diagnosis” – but earlier these percentages seem to refer to COVID status for all patients (Figure 3)?
- Page 32 discusses a subgroup analysis separating NOFF and non-NOFF mortality, demonstrated in Table 4. I am unable to find these results in Table 4 (or anywhere else).
- Could the increased mortality during the pandemic partly be due to selection of more severe cases? That is, could the decrease in referrals and surgical interventions indicate a higher threshold for treatment (due to a redistribution of hospital resources during the pandemic), in turn making the two cohorts different in terms of severity? Table 2 indicates that the COVID population on average is slightly older, so maybe this population was frailer than in “normal” years?

	The manuscript is generally well-written, but could benefit from some language editing. See for instance the multiple mentions of PR instead of RR on pages 23-24.
--	--

VERSION 1 – AUTHOR RESPONSE

Reviewer: 1

Dr. Carlos Prada, McMaster University

Comments to the Author:

Dear authors:

Thank you for submitting your work to the journal.

This multi-centre observational study aim to assess the impact of the COVID-19 pandemic on orthopaedic trauma with respect to referrals, operative caseload and mortality during COVID-19 first wave "peak" in the UK during 2020.

Main comments:

1. Relevance: I have to start saying that I acknowledge the effort behind this initiative. However, the results are not a surprise and do not add significant information to the current discussion and what has been published recently on this topic. Despite being a big study with robust collected data the novelty of this manuscript is minor.

Thank you for your opinion, but this is not necessarily true. If the reviewer wishes to make such a statement, may we kindly ask for references to suggest that a similar study has been conducted in the UK which is multi-centre and with that many patient encounters during any of the waves of the pandemic in the past 18 months for orthopaedic trauma. It may not be a surprise to the reviewer, but this is study certainly contributes to literature by directly observing effect on mortality.

2. Results: In more than one opportunity in the results section the authors provide explanations to certain factors/outcomes. I suggest to remove those paragraphs and address those aspects in the discussion. The result section should just show a summary of the collected data in a friendly way to the readers without judgment or appraisal.

Done – explanations have been moved to discussion section

3. Discussion:

A. The authors suggest that the increase in the incidence of neck femoral fractures could have been explained by prodromal symptoms of COVID-19, without clarifying what was the percentage of patients with these fractures that were COVID-19 (+) to weight the relevance of this argument. In addition, it seems more likely that given that elderly people sometimes need help at their home and could have lacked of the assisstance they used to have because of the national lock-down, so the need to perform tasks by themselves coud have put them at a higher risk.

Done – added in discussion as well as incorporating your suggestion

B. What are the causes behind the increase in the overall mortality in the authors opinion? It is unclear to me what factors they attribute to be the cause(s) behind this difference, specially in the COVID-19 (-) population. This is a key aspect of the manuscript and definitely needs to be developed more exhaustively.

Done – added in the 'observation of hip fractures' section under discussion. This may not be the right place to delve into reasons exhaustively since this study is looking at the overall orthopaedic trauma mortality but we have referenced focussed studies on hip fractures during COVID study that we were also a part of for further information.

Minor comments:

- “Golden peak” concept: No references are supporting the decision of declaring this 6-week period as the peak. Epidemiological COVID-19 data from those weeks should be added and referenced to give the readers perspective of the magnitude of the first wave in the included territories to help with external validity of the study.

Done – added justification with referencing from Office of National Statistics

- Statistics: I suggest to check that the ASA grade had a normal distribution. Commonly, this variable is non-normally distributed. If that’s the case, 95% CI are not preferred for the table showing ASA grade. 95% CI showing 2-2 is not informative so, as this variable is probably skewed, median and interquartile ranges might be needed instead.

Done – and changed method of analysis in ‘statistical analysis’

Thank you again for submitting your work.

Best regards,

** ** *

Reviewer: 2

Dr. Eyvind Ohm, Norwegian Institute of Public Health

Comments to the Author:

Thank you for the opportunity to review this interesting manuscript, which significantly adds to our knowledge about the impact of COVID-19 on trauma care and orthopaedic workload. Specifically, the authors document a reduction in trauma referrals and surgical interventions during the early phase of the pandemic, combined with an increase in mortality for this patient population. Major strengths include the assessment of mortality, and details about COVID status of trauma patients. However, I do have some comments about the manuscript. The Results section needs some work, and some text passages seem oddly placed (see below for details).

Abstract: under “Setting” it is claimed that hospitals from seven cities were recruited. However, from the description of the contributing hospitals (pages 3-9), I count seven hospitals from six cities (two hospitals are located in London). Further, I would have liked to see a list of these hospitals in the main text (e.g., in Methods), since it is claimed (without documentation) that this study is representative of the whole UK.

The hospitals have been highlighted in the title page with contributors from each centre. We have mentioned the six cities in the methodology as per your request.

Introduction: It seems redundant to describe the outcomes in the Introduction, since this information is repeated in Methods (where such details belong).

Done – outcomes moved to methods

Methods:

- The date given for the end of the study period in 2020 (page 16) is wrong (should be Tuesday the 28th of April, not the 31st)

Done – thank you for spotting that – changed

- Inclusion/exclusion criteria: Elective procedures were excluded, yet “patients listed for an operation prior to time period of data collection were included” – this is confusing to me. The latter operations seem elective in nature, since these traumas could be postponed until a later date.

So elective trauma is almost an oxymoron and not all trauma needs to be operated on immediately. So if the trauma can wait, it can be operated on later – usually within 2-3 week window which is still considered as acute/trauma surgery. The notion of elective in orthopaedics refers to surgery planned months in advance such as arthroplasty due to arthritis for instance. This has been clarified in the 'inclusion criteria'.

- COVID status: are the three groups swabbed mutually exclusive? Tests on those swabbed due to presence of documented symptoms (group 2) could result in both negative (group 3) or positive (group 4) results. Or?

This has been clarified as two distinct groups only: swabbed or not swabbed

Results:

- The number of tables/figures must surely exceed the maximum allowed for this journal? Some figures are not very informative (e.g., Figure 6) and could be summarized in the text.

Thank you for the suggestion. This has been brought up prior to the submission to the reviewers. This will be left up to the editors as to which figures will be left in the text body or moved to appendix. For convenience and to reduce confusion, figures 3 and 4 have also been combined to reduce the number of total figures.

- I would have liked to see a sentence or two describing the main results from Table 2, documenting the overall change in referrals/surgical interventions (as done in the Abstract). Likewise, the results from the main descriptive analyses shown in Figures 1 and 2 could briefly be summarized to get an overall picture of changes witnessed during the pandemic.

Done – figures 1 and 2 are discussed in depth in the discussion and would be difficult to summarise in a couple of sentences.

- Figure 1: why is “infection” both an injury type and mechanism? Seems redundant to include twice in the same figure.

Infection is a subcategory for type and mechanism of injury which are not direct comparisons but rather mutually exclusive.

- In Figure 1, the bar to the left (in dark) refers to the pre-covid period while the bar to the right (in grey) refers to the covid period. In Figure 2 this is the other way around, appearing backwards in time.

Done - Changed

- COVID status: information about testing (first sentence on page 21) belongs in Methods.

Done - Moved

- I found Figure 4 (“COVID status of all mortalities”), and the text on page 25, difficult to understand. 6-week mortality is straightforward, but what does 6% and 8.1% for COVID positive during the COVID period mean? Is this the percentage of dead patients with a positive COVID result, or the percentage of patients with COVID who died? Or something else? Presentation of these percentages was confusing to me.

Done – this has been simplified by combining figures 3 and 4 together to reduce number of figures and help with direct comparison.

- Table 3 (and the text introducing this table on page 23): ratios here refer to comparisons between pre-COVID vs. COVID periods, but this is not made explicit in the caption/title.

Done – caption altered.

- Some of the text on pages 23-24 sounds like discussion (“...perhaps to create a closed circuit...”, “...this could have skewed the data...”, “...as a consequence of all elective operations

being suspended...”, “This could be due to..”). I would rephrase this and/or move these speculations to the Discussion.

Done – rephrased and moved to discussion section

- Figure 7a/b: I found these figures difficult to read. Bar charts may be more intuitive to get the message across. Further, much of the text regarding comorbidities focuses on comparing results from other studies (page 27) and belongs in the Discussion.

Done – changed to bar charts to make it easier to compare – the text has been moved to discussion.

- Survival analyses: how can mortality rates for operative cases double during the COVID period (Table 3), but survival probability be lowest pre-COVID?

Table 3 looks at the risk/prevalence and odds ratios which is not as simple as saying that mortality has ‘doubled’ – this has to be linked to the context of the ratios. The mortality rates are absolute and measured within a 6-week period. The survival probability adds to the trends of mortality relative to the time period at various time points. Hence, although the absolute number of mortalities doubled during COVID, the number of days until death was lower in the pre-COVID era is seen in figures 7a-b. This is discussed in the mortality section under discussion as well as other possible reasons for the trend (e.g. ring-fenced beds with closer medical/nursing supervision etc.)

- The last two paragraphs of Results (pages 29-30) sound more like background/discussion.

Done – moved to discussion

Discussion:

- Mortality: “...and a third (29-32%) of those deaths had a positive COVID-19 diagnosis” – but earlier these percentages seem to refer to COVID status for all patients (Figure 3)?

This has been reworded to clarify further – thank you for pointing this out

- Page 32 discusses a subgroup analysis separating NOFF and non-NOFF mortality, demonstrated in Table 4. I am unable to find these results in Table 4 (or anywhere else).

You are correct – this sentence has been removed since a section in the discussion has been allocated specifically for NOFF mortalities – thank you for pointing this out.

- Could the increased mortality during the pandemic partly be due to selection of more severe cases? That is, could the decrease in referrals and surgical interventions indicate a higher threshold for treatment (due to a redistribution of hospital resources during the pandemic), in turn making the two cohorts different in terms of severity? Table 2 indicates that the COVID population on average is slightly older, so maybe this population was frailer than in “normal” years?

Done - this point has been added in discussion too.

The manuscript is generally well-written, but could benefit from some language editing. See for instance the multiple mentions of PR instead of RR on pages 23-24.

PR and RR are synonymous which was mentioned in the ‘Risk (or prevalence) and odds ratios’ – hence this was done on purpose

VERSION 2 – REVIEW

REVIEWER	Prada, Carlos McMaster University, Orthopaedics
REVIEW RETURNED	01-Nov-2021

GENERAL COMMENTS	Dear authors, Thank you for submitting your revised version of this manuscript. I believe the authors tackled successfully our comments. As per the authors request in their response to reviewers comments, I added below a couple of references of articles assessing similar outcomes in both, multi-centre and single-centre studies. I acknowledge and appreciate the effort behind this multi-centre initiative but the results, as I declared before, do not add much more than more robust data to what has been published and novelty is one aspect that we as reviewers are asked to assess. - Karayiannis PN, Roberts V, Cassidy R, Mayne AIW, McAuley D, Milligan DJ, Diamond O. 30-day mortality following trauma and orthopaedic surgery during the peak of the COVID-19 pandemic: a multicentre regional analysis of 484 patients. Bone Jt Open. 2020 Nov 2;1(7):392-397. doi: 10.1302/2633-1462.17.BJO-2020-0075.R1. PMID: 33215129; PMCID: PMC7659654. - Balakumar B, Nandra RS, Woffenden H, Atkin B, Mahmood A, Cooper G, Cooper J, Hindle P. Mortality risk of surgically managing orthopaedic trauma during the COVID-19 pandemic. Bone Jt Open. 2021 May;2(5):330-336. doi: 10.1302/2633-1462.25.BJO-2020-0189.R1. PMID: 34027674; PMCID: PMC8168546. - Greenhalgh M, Dupley L, Unsworth R, Boden R. Where did all the trauma go? A rapid review of the demands on orthopaedic services at a UK Major Trauma Centre during the COVID-19 pandemic. Int J Clin Pract. 2021 Mar;75(3):e13690. doi: 10.1111/ijcp.13690. Epub 2020 Sep 14. PMID: 32852851; PMCID: PMC7460967. Thank you for submitting your article to this journal.
--

REVIEWER	Ohm, Eyvind Norwegian Institute of Public Health, Department of Health and Inequality
REVIEW RETURNED	19-Oct-2021

GENERAL COMMENTS	Thank you for this revised manuscript, in which the authors have answered most of my queries. However, I have some comments, mainly concerning the Results section, which I still believe needs some work. 1. While the authors have moved passages that included explanations and interpretation to the Discussion (as requested), I feel they now have gone too far, such that the Results section (in places) lacks a proper presentation of the results. For instance, there is no attempt to summarize covid status among patients in 2020 (page 14) or to describe what the Kaplan-Meier plots show (page 18). The Results section should include a brief description/summary of the results (in words), not just a reference to some figure or table. I do not think that simply moving whole paragraphs in the previous version (which included both presentation of results and interpretation) from Results to Discussion is satisfactory (as some of this text belongs in Results). Moreover, as a consequence of these changes, many findings are now instead introduced in the Discussion section (e.g., ORs of road traffic accidents/sporting injuries/infection and PRs of open reduction/internal fixation/removal of metalwork/foreign bodies/dynamic hip screw fixation on page 20), making the Discussion seem overly detailed. 2. Generally, I find the Results section somewhat “messy” and not very reader-friendly, with multiple analyses (some of which are not
--

	summarized, see point 1) and a large number of tables/figures to keep track of. It feels as though this study tries to cover too much. Are all analyses in fact crucial to convey? After reading this section several times, I still do not feel I have a “grasp” of all the results. I would still like to see a reduction in the number of figures and a more intuitive organization when presenting the results. For instance, could one possibility be to split the subheading “Morbidity and Mortality”, since many of these analyses seem to apply only to patients who died (e.g., Table 4, Figures 5-7)?
--	---

VERSION 2 – AUTHOR RESPONSE

Reviewer: 1

Dr. Carlos Prada, McMaster University

Comments to the Author:

Dear authors:

Thank you for submitting your work to the journal.

This multi-centre observational study aim to assess the impact of the COVID-19 pandemic on orthopaedic trauma with respect to referrals, operative caseload and mortality during COVID-19 first wave "peak" in the UK during 2020.

Main comments:

1. Relevance: I have to start saying that I acknowledge the effort behind this initiative. However, the results are not a surprise and do not add significant information to the current discussion and what has been published recently on this topic. Despite being a big study with robust collected data the novelty of this manuscript is minor.

Thank you for your opinion, but this is not necessarily true. If the reviewer wishes to make such a statement, may we kindly ask for references to suggest that a similar study has been conducted in the UK which is multi-centre and with that many patient encounters during any of the waves of the pandemic in the past 18 months for orthopaedic trauma. It may not be a surprise to the reviewer, but this is study certainly contributes to literature by directly observing effect on mortality.

2. Results: In more than one opportunity in the results section the authors provide explanations to certain factors/outcomes. I suggest to remove those paragraphs and address those aspects in the discussion. The result section should just show a summary of the collected data in a friendly way to the readers without judgment or appraisal.

Done – explanations have been moved to discussion section

3. Discussion:

A. The authors suggest that the increase in the incidence of neck femoral fractures could have been explained by prodromal symptoms of COVID-19, without clarifying what was the percentage of patients with these fractures that were COVID-19 (+) to weight the relevance of this argument. In addition, it seems more likely that given that elderly people sometimes need help at their home and could have lacked of the assistance they used to have because of the national lock-down, so the need to perform tasks by themselves could have put them at a higher risk.

Done – added in discussion as well as incorporating your suggestion

B. What are the causes behind the increase in the overall mortality in the authors opinion? It is unclear to me what factors they attribute to be the cause(s) behind this difference, specially in the COVID-19 (-) population. This is a key aspect of the manuscript and definitely needs to be developed more exhaustively.

Done – added in the 'observation of hip fractures' section under discussion. This may not be the right place to delve into reasons exhaustively since this study is looking at the overall orthopaedic trauma mortality but we have referenced focussed studies on hip fractures during COVID study that we were also a part of for further information.

Minor comments:

- "Golden peak" concept: No references are supporting the decision of declaring this 6-week period as the peak. Epidemiological COVID-19 data from those weeks should be added and referenced to give the readers perspective of the magnitude of the first wave in the included territories to help with external validity of the study.

Done – added justification with referencing from Office of National Statistics

- Statistics: I suggest to check that the ASA grade had a normal distribution. Commonly, this variable is non-normally distributed. If that's the case, 95% CI are not preferred for the table showing ASA grade. 95% CI showing 2-2 is not informative so, as this variable is probably skewed, median and interquartile ranges might be needed instead.

Done – and changed method of analysis in 'statistical analysis'

Thank you again for submitting your work.

Best regards,

** ** **

Reviewer: 2

Dr. Eyvind Ohm, Norwegian Institute of Public Health

Comments to the Author:

Thank you for the opportunity to review this interesting manuscript, which significantly adds to our knowledge about the impact of COVID-19 on trauma care and orthopaedic workload. Specifically, the authors document a reduction in trauma referrals and surgical interventions during the early phase of the pandemic, combined with an increase in mortality for this patient population. Major strengths include the assessment of mortality, and details about COVID status of trauma patients. However, I do have some comments about the manuscript. The Results section needs some work, and some text passages seem oddly placed (see below for details).

Abstract: under "Setting" it is claimed that hospitals from seven cities were recruited. However, from the description of the contributing hospitals (pages 3-9), I count seven hospitals from six cities (two hospitals are located in London). Further, I would have liked to see a list of these hospitals in the main text (e.g., in Methods), since it is claimed (without documentation) that this study is representative of the whole UK.

The hospitals have been highlighted in the title page with contributors from each centre. We have mentioned the six cities in the methodology as per your request.

Introduction: It seems redundant to describe the outcomes in the Introduction, since this information is repeated in Methods (where such details belong).

Done – outcomes moved to methods

Methods:

- The date given for the end of the study period in 2020 (page 16) is wrong (should be Tuesday the 28th of April, not the 31st)

Done – thank you for spotting that – changed

- Inclusion/exclusion criteria: Elective procedures were excluded, yet "patients listed for an operation prior to time period of data collection were included" – this is confusing to me. The latter operations seem elective in nature, since these traumas could be postponed until a later date.

So elective trauma is almost an oxymoron and not all trauma needs to be operated on immediately. So if the trauma can wait, it can be operated on later – usually within 2-3 week window which is still considered as acute/trauma surgery. The notion of elective in orthopaedics refers to surgery planned months in advance such as arthroplasty due to arthritis for instance. This has been clarified in the 'inclusion criteria'.

- COVID status: are the three groups swabbed mutually exclusive? Tests on those swabbed due to presence of documented symptoms (group 2) could result in both negative (group 3) or positive (group 4) results. Or?

This has been clarified as two distinct groups only: swabbed or not swabbed

Results:

- The number of tables/figures must surely exceed the maximum allowed for this journal? Some figures are not very informative (e.g., Figure 6) and could be summarized in the text.

Thank you for the suggestion. This has been brought up prior to the submission to the reviewers. This will be left up to the editors as to which figures will be left in the text body or moved to appendix. For convenience and to reduce confusion, figures 3 and 4 have also been combined to reduce the number of total figures.

- I would have liked to see a sentence or two describing the main results from Table 2, documenting the overall change in referrals/surgical interventions (as done in the Abstract). Likewise, the results from the main descriptive analyses shown in Figures 1 and 2 could briefly be summarized to get an overall picture of changes witnessed during the pandemic.

Done – figures 1 and 2 are discussed in depth in the discussion and would be difficult to summarise in a couple of sentences.

- Figure 1: why is “infection” both an injury type and mechanism? Seems redundant to include twice in the same figure.

Infection is a subcategory for type and mechanism of injury which are not direct comparisons but rather mutually exclusive.

- In Figure 1, the bar to the left (in dark) refers to the pre-covid period while the bar to the right (in grey) refers to the covid period. In Figure 2 this is the other way around, appearing backwards in time.

Done - Changed

- COVID status: information about testing (first sentence on page 21) belongs in Methods.

Done - Moved

- I found Figure 4 (“COVID status of all mortalities”), and the text on page 25, difficult to understand. 6-week mortality is straightforward, but what does 6% and 8.1% for COVID positive during the COVID period mean? Is this the percentage of dead patients with a positive COVID result,

or the percentage of patients with COVID who died? Or something else? Presentation of these percentages was confusing to me.

Done – this has been simplified by combining figures 3 and 4 together to reduce number of figures and help with direct comparison.

- Table 3 (and the text introducing this table on page 23): ratios here refer to comparisons between pre-COVID vs. COVID periods, but this is not made explicit in the caption/title.

Done – caption altered.

- Some of the text on pages 23-24 sounds like discussion (“...perhaps to create a closed circuit...”, “...this could have skewed the data...”, “...as a consequence of all elective operations being suspended...”, “This could be due to..”). I would rephrase this and/or move these speculations to the Discussion.

Done – rephrased and moved to discussion section

- Figure 7a/b: I found these figures difficult to read. Bar charts may be more intuitive to get the message across. Further, much of the text regarding comorbidities focuses on comparing results from other studies (page 27) and belongs in the Discussion.

Done – changed to bar charts to make it easier to compare – the text has been moved to discussion.

- Survival analyses: how can mortality rates for operative cases double during the COVID period (Table 3), but survival probability be lowest pre-COVID?

Table 3 looks at the risk/prevalence and odds ratios which is not as simple as saying that mortality has ‘doubled’ – this has to be linked to the context of the ratios. The mortality rates are absolute and measured within a 6-week period. The survival probability adds to the trends of mortality relative to the time period at various time points. Hence, although the absolute number of mortalities doubled during COVID, the number of days until death was lower in the pre-COVID era is seen in figures 7a-b. This is discussed in the mortality section under discussion as well as other possible reasons for the trend (e.g. ring-fenced beds with closer medical/nursing supervision etc.)

- The last two paragraphs of Results (pages 29-30) sound more like background/discussion.

Done – moved to discussion

Discussion:

- Mortality: “...and a third (29-32%) of those deaths had a positive COVID-19 diagnosis” – but earlier these percentages seem to refer to COVID status for all patients (Figure 3)?

This has been reworded to clarify further – thank you for pointing this out

- Page 32 discusses a subgroup analysis separating NOFF and non-NOFF mortality, demonstrated in Table 4. I am unable to find these results in Table 4 (or anywhere else).

You are correct – this sentence has been removed since a section in the discussion has been allocated specifically for NOFF mortalities – thank you for pointing this out.

- Could the increased mortality during the pandemic partly be due to selection of more severe cases? That is, could the decrease in referrals and surgical interventions indicate a higher threshold for treatment (due to a redistribution of hospital resources during the pandemic), in turn making the two cohorts different in terms of severity? Table 2 indicates that the COVID population on average is slightly older, so maybe this population was frailer than in “normal” years?

Done - this point has been added in discussion too.

The manuscript is generally well-written, but could benefit from some language editing. See for instance the multiple mentions of PR instead of RR on pages 23-24.

PR and RR are synonymous which was mentioned in the ‘Risk (or prevalence) and odds ratios’ – hence this was done on purpose